# Aza-BODIPY: A New Vector for Enhanced Theranostic Boron Neutron Capture Therapy Applications

**DOI:** 10.3390/cells9091953

**Published:** 2020-08-25

**Authors:** Ghadir Kalot, Amélie Godard, Benoît Busser, Jacques Pliquett, Mans Broekgaarden, Vincent Motto-Ros, Karl David Wegner, Ute Resch-Genger, Ulli Köster, Franck Denat, Jean-Luc Coll, Ewen Bodio, Christine Goze, Lucie Sancey

**Affiliations:** 1Institute for Advanced Biosciences, UGA INSERM U1209 CNRS UMR5309, 38700 La Tronche, France; Ghadir.kalot@univ-grenoble-alpes.fr (G.K.); bbusser@chu-grenoble.fr (B.B.); mans.broekgaarden@univ-grenoble-alpes.fr (M.B.); jean-luc.coll@univ-grenoble-alpes.fr (J.-L.C.); 2Institut de Chimie Moléculaire de l’Université de Bourgogne, ICMUB CNRS, UMR 6302, Université Bourgogne Franche-Comté, 21078 Dijon, France; amelie_godard@etu.u-bourgogne.fr (A.G.); jacques.pliquett@googlemail.com (J.P.); franck.denat@u-bourgogne.fr (F.D.); 3Grenoble Alpes University Hospital, 38043 Grenoble, France; 4Institut Lumière Matière UMR 5306, Université Lyon 1-CNRS, Université de Lyon, 69622 Villeurbanne, France; vincent.motto-ros@univ-lyon1.fr; 5Division Biophotonics, Federal Institute for Materials Research and Testing (BAM), Richard-Willstaetter-Str. 11, 12489 Berlin, Germany; karl-david.wegner@bam.de (K.D.W.); ute.resch@bam.de (U.R.-G.); 6Institut Laue Langevin, 38042 Grenoble, France; koester@ill.fr

**Keywords:** aza-BODIPY, BNCT, in ovo model, theranostic, boron compound, ^10^B-BSH, optical imaging, NIR-I, SWIR

## Abstract

Boron neutron capture therapy (BNCT) is a radiotherapeutic modality based on the nuclear capture of slow neutrons by stable ^10^B atoms followed by charged particle emission that inducing extensive damage on a very localized level (<10 μm). To be efficient, a sufficient amount of ^10^B should accumulate in the tumor area while being almost cleared from the normal surroundings. A water-soluble aza-boron-dipyrromethene dyes (BODIPY) fluorophore was reported to strongly accumulate in the tumor area with high and BNCT compatible Tumor/Healthy Tissue ratios. The clinically used ^10^B-BSH (sodium borocaptate) was coupled to the water-soluble aza-BODIPY platform for enhanced ^10^B-BSH tumor vectorization. We demonstrated a strong uptake of the compound in tumor cells and determined its biodistribution in mice-bearing tumors. A model of chorioallantoic membrane-bearing glioblastoma xenograft was developed to evidence the BNCT potential of such compound, by subjecting it to slow neutrons. We demonstrated the tumor accumulation of the compound in real-time using optical imaging and ex vivo using elemental imaging based on laser-induced breakdown spectroscopy. The tumor growth was significantly reduced as compared to BNCT with ^10^B-BSH. Altogether, the fluorescent aza-BODIPY/^10^B-BSH compound is able to vectorize and image the ^10^B-BSH in the tumor area, increasing its theranostic potential for efficient approach of BNCT.

## 1. Introduction

Boron neutron capture therapy (BNCT) is a cancer treatment modality based on the vectorization of ^10^B-rich compounds in tumor tissues before neutron exposure to selectively destroy cancer cells. Under low-energy neutron irradiation, the stable ^10^B atoms may capture neutrons producing energetic alpha and ^7^Li particles. The generated high-linear energy transfer (LET) particles have a cell killing effect within a 10 µm-range [1,2]. Such type of cancer treatment may not only spare the surrounding healthy tissues but may be efficient to treat recurrent, or radioresistant to conventional X-ray photon therapy tumors [3,4,5,6].

BNCT has obtained promising clinical results for several pathologies as head and neck tumors including recurrent pathologies [3,5,7], malignant brain tumors [8,9], and malignant melanoma [10,11]. Regardless of these reports, there are still several limitations. The first limitation of this treatment modality meanwhile overcome was the lack of hospital neutron sources that limited the clinical practice to research nuclear reactor sites. Recent developments have permitted the installation of accelerator-based neutron sources [12,13], opening new perspectives to BNCT. In the last 4 decades, only two compounds have been used as ^10^B-sources for BNCT in patients: sodium mercaptoundecahydrododecaborate (also called sodium borocaptate, Na_2_^10^B_12_H_11_SH; Na_2_^10^BSH; or BSH) and L-*p*-boronophenylalanine (L-^10^BPA). While not used in clinical trials, GB-10 ([closo-B_10_H_10_]^2−^, dodecahydrododecaborate) is also an FDA approved molecule [14]. Despite their clinical use, both compounds do not fulfill all the required criteria. In particular, to be successful, 20 to 50 µg of ^10^B per gram of tumor is necessary, with a tumor-to-normal tissue and tumor-to-blood ratio > 3:1 [15].

To deliver boron-containing compounds in tumors, various low molecular weight molecules have been developed for preclinical research including boron clusters [16,17], amino acid derivatives [18], RGD-BSH conjugates for integrin α_v_β_3_ targeting [19], and several other specific targeted systems [20,21,22]. The clinical application of BNCT requires the tracking of the compound in vivo to predict the ^10^B concentration in the tumor and surrounding non-tumor tissues. Blood samples are required for indirect ^10^B tumor amount estimation [23,24] and adjustment of the treatment planning system that permits to deliver the optimal neutron exposure time and to determine the dose delivered to the patient. Positron emission tomography (PET) or single-photon emission computed tomography (SPECT), which are highly sensitive nuclear imaging techniques, are non-invasive quantification procedures used to track and quantify boron carriers in vivo, for personalized treatment protocols.

For preclinical studies, optical imaging (OI) can be also used [25,26]. Among the different optical probes, boron-dipyrromethene dyes (BODIPYs, Figure 1a) are very versatile organic fluorophores with tunable optical properties from the visible to the NIR-I (Near Infrared) and SWIR (Short Wave Infrared) optical windows [27,28,29], which display excellent photophysical properties controlled by substitution pattern conferring a strong interest for these compounds as theranostic tools [30,31]. Moreover, the fluorine atoms on the boron could be substituted by acetylides bearing ammonium groups for engrafting functional groups to increase the water-solubility and to conjugate them with chelates for OI/PET, OI/SPECT bimodal imaging [32,33,34,35], and therapeutic moieties [31,36,37]. BODIPY fluorophores are also interesting for BNCT, as they possess a boron atom. However, the presence of only a single boron atom in the BODIPY core and the limited number of commercially available ^10^B-boron reagents make it difficult to synthesize enriched ^10^B-BODIPYs. For this reason, BODIPY derivatives reported for a BNCT use implied their conjugation to a ^10^B-boron cluster [38,39,40,41]. Up-to-now, none of these studies described either in vivo BNCT investigations or OI distribution studies due to the fact that BODIPYs emit in the visible region of the light spectrum while preclinical OI studies require fluorescence emission in the NIR-I and, if possible, in the NIR-II/SWIR optical window. To bridge this gap, we decided to tether a ^10^B-BSH on a NIR-emitting BODIPY derivative named aza-BODIPY (Figure 1b), thereby creating a theranostic system gathering the advantages of both aza-BODIPY and ^10^B-BSH components.

We recently reported the use of a *B*-functionalized aza-BODIPY emitting in the SWIR region and accumulating in tumors via EPR effect (Enhanced permeability and retention effect) with high and prolonged Tumor/Healthy Muscle ratio (between 5 and 30, from 24 to 168 h post-injection) without the need of tethering vector [29]. SWIR fluorophores enable to reach a penetration depth of 1 to 10 mm with increased resolution of the images as compared to those obtained with NIR-I probes [42,43]. Here, such aza-BODIPY was used to vectorize the small ^10^B-BSH molecule at the tumor site while benefitting from the imaging properties of the fluorophore for theranostic applications. First, the tumor cell loading capacity of the compound was demonstrated using original microscopy settings and its in vitro BNCT efficacy. Then, its distribution and behavior in tumor-bearing mice were described. For BNCT experiments performed on animals, the chorioallantoic membrane (CAM) model in fertilized chicken eggs was used to establish vascularized glioblastoma in vivo and evaluated the aza-BODIPY/^10^B-BSH conjugate tumor uptake with optical imaging and laser-induced breakdown spectroscopy [44,45,46,47]. Finally, a neutron beam was used to evaluate the expected toxicity of the neutron-exposed aza-BODIPY/^10^B-BSH compounds and demonstrated their strong potential as efficient theranostic boron-vectors for promising BNCT applications.

## 2. Materials and Methods

### 2.1. Synthesis and Characterization of Compounds

Detailed syntheses and analyses—NMR, mass-analyses (the calculations are based on the mass of the most abundant isotopologue, unless otherwise specified), and HPLC-MS—can be found in ESI and previously reported studies [29,48]. (^1^H, ^11^B, ^13^C, ^19^F)-NMR spectra were recorded at 300 K on Bruker 500 Avance III or 600 Avance III spectrometers. Chemical shifts are given relative to TMS (tetramethylsilane) (^1^H, ^13^C), BF_3_*Et_2_O (^11^B), CFCl_3_ (^19^F), and were referenced to the residual solvent signal. High-resolution mass spectra (HR-MS) were recorded on a Thermo LTQ Orbitrap XL ESIMS spectrometer. NMR and mass-analyses were performed at the “Plateforme d’Analyse Chimique et de Synthèse Moléculaire de l’Université de Bourgogne” (PACSMUB). The photophysical characterization has been performed as described previously [29] using UV-Vis-NIR spectrophotometer Cary5000 between 300 and 1200 nm. Steady-state photoluminescence spectra were measured from 700 to 1500 nm with a calibrated FSP 920 (Edinburgh Instruments, Edinburgh, UK) spectrofluorometer equipped with a nitrogen-cooled PMT R5509P.

### 2.2. Cell Lines and Culture

Human glioblastoma astrocytoma cell lines, U-87 MG and U-251 MG, were obtained from the European Collection of Authenticated Cell Cultures (ECACC).

U-87 MG and U-251 MG cell lines were cultured in a 37 °C humidified environment containing 5% CO_2_ in DMEM media supplemented with 10% heat-inactivated fetal bovine serum, and for U-87 MG cell cultures 1% non-essential amino acids. A total of 10,000 cells were plated into 4-well chambered cover glass Nunc^TM^ Lab-Tek^TM^ II (Roskilde, Denmark) for 48 h to be used for fluorescence microscopy.

### 2.3. Fluorescence Microscopy

Previously prepared 2D cell cultures were kept at 37 °C, 5% CO_2_ and incubated with 10 µM aza-SWIR-BSH-01 compound solution diluted in cell culture medium. Fluorescence microscopy images were acquired using a confocal laser-scanning microscope (LSM 710 Carl Zeiss, Jena, Germany) in combi mode. Plan apochromat 63× in oil objective was used. Hoechst 33,342 was used to counterstain the cell nuclei (1 µM). Optimal fluorescent signal would be obtained with a 680 nm excitation laser, and a collection wavelength between 800–1200 nm would be required. However, such settings are not available in our current microscopy system. Therefore, the fluorescent signal of the water-soluble aza-BODIPY was collected after excitation with 1.5% 633 nm laser (pinhole aperture 200 µm, gain 1000) using a LP736 filter, in APD (avalanche photodiode) mode.

Images were processed using ImageJ software. The experiments were performed at the MicroCell (Optical Microscopy-Cell Imaging) platform, IAB Grenoble.

### 2.4. In Vivo Imaging Experiments

All animal experiments were performed in accordance with the Institutional Animal Care and Use Committee at Grenoble Alpes University. These experiments were also approved by the Animal Ethics Committee of the French Ministry, under the agreement number APAFIS #8782. The experiments were performed at the Optimal (Small Animal Imaging Platform) platform, IAB Grenoble.

U-87 MG cancer cells (5 million cells per 100 µL PBS) were injected subcutaneously on the right flank of female NMRI nude mice (6-8-week-old) (Janvier Labs, Le Genest-Saint Isle, France). After tumor growth (~two weeks), 6 mice were anesthetized (air/isoflurane 4% for induction and 2% thereafter) and injected intravenously in the tail vein with 200 µL of aza-SWIR-BSH-01 (600 µM in PBS). Whole-body NIR fluorescence images were acquired before and 2, 5, 24, and 48 h post-administration. Three mice were euthanized at 24 and 48 h, respectively, and their organs were sampled for ex vivo fluorescence imaging. Acquired images were analyzed using ImageJ software. Semi-quantitative data were obtained by drawing regions of interest (ROI) around the organs. The fluorescence imaging was performed using a Pear Trilogy LI-COR system with a laser excitation source of 785 nm and a CCD (charge couple device) collecting fluorescence > 820 nm.

### 2.5. Neutron Exposure at the Institut Laue-Langevin

#### 2.5.1. Neutron Beam Characteristics

Neutron irradiation experiments were performed at the Institut Laue-Langevin (ILL) (Grenoble, France). The beam line PF1b [49] provides cold neutrons that were collimated through a 3 m long system to obtain a final circular beam of 2 cm diameter. Before the experiment, the thermal neutron capture equivalent flux was measured by activation of thin Au foils, at 2.85 × 10^9^ cm^−2^ s^−1^. For cell irradiations, this cold neutron beam provides results equivalent to a thermal neutron irradiation as explained in detail in Pedrosa-Riviera et al. [50], but for egg irradiations, a significant attenuation towards deeper layers has to be considered.

#### 2.5.2. BNCT Experiment on Cells

Exponentially growing glioblastoma cancer cells (U251 MG/U87 MG cells) were incubated with media alone, media containing BSH, or media containing aza-SWIR-BSH-01, at a ^10^B concentration of 40 µg/mL. Cells incubated with media alone were used as a control. After 2 h, the cells (in each incubation condition) were washed, collected, and divided into 4 quartz cuvettes (Hellma, 110-QS, 1-mm layer thickness). These cuvettes were kept as control or exposed to the neutron beam for 1, 5, and 10 min, respectively. Following the neutron irradiation, cells were counted, diluted, and re-plated in T25 flasks in triplicates (500 cells/flask) for the colony formation assay. When 128-cell-colonies were formed in the control condition, the flasks were washed with PBS, fixed with 4% formol (U-87 MG) (for 15 min) or with 4% glutaraldehyde (U-251 MG) (for 5 min), stained with methylene blue (for 15–30 min), and dried. Finally, the colonies were counted and the results were normalized to the control condition in each cell line and treatment condition.

#### 2.5.3. In Ovo BNCT Experiment

According to the French and European regulations, no ethical approval is needed for the scientific experimentations using oviparous embryos. This model was developed to restrict the use of animals and to facilitate BNCT experiments, according to the following procedure: fertilized white leghorn chicken eggs (Couvoir de Cerveloup, Vourey, France) were incubated at 38 °C with 60% relative humidity. At day-3 of chicken embryo development, 3 mL albumin was removed from the eggs and a small window was made into the eggshell above the chorioallantoic membrane (CAM). At day 7, the CAM was gently lacerated and 5 × 10^6^ pelleted glioma cells mixed with 30 µL of matrigel (Growth Factor Reduced (GFR) Basement Membrane Matrix 354,230, Corning^®^ Matrigel^®^ (Wiesbaden, Germany)) were deposited on the lacerated region. At day 10, 100 µL of ^10^B-containing formulations (^10^B-BSH or aza-SWIR-BSH-01) were added on top of the grown tumors (1.35 µg of ^10^B/egg) 1 h prior to irradiation, untreated eggs served as a control. The eggs were then exposed to neutron beam for 1 h, split in 2 sessions of 30 min to expose each tumor on both sides (n ≤ 3/condition). Following neutron exposure, the eggs were re-incubated for additional 6 days after which they were terminated and the tumors were carefully collected, weighed, and analyzed.

#### 2.5.4. Statistical Analysis

In vitro data passed the Shapiro-Wilk normality test and were analyzed using a One-way ANOVA and a Sidak’s test for multiple comparisons (Graphpad Prism 7.0 (La Jolla, CA, USA)). CAM data passed the Shapiro-Wilk normality test and the indicated treatment groups were compared using a student’s *t*-test.

### 2.6. Elemental Imaging Using Laser-Induced Breakdown Spectroscopy (LIBS)

Frozen tumor samples were cut into 7 μm-thick sections before mounting onto plastic slides. The samples were analyzed with a LIBS system to determine their elemental composition, i.e., boron (B (I) 208.8 nm) and phosphorus (P (I) 214.9 nm) elements in our case. The homemade LIBS setup was based on an optical microscope that combined a LIBS laser injection line, a standard optical-imaging apparatus, and a three-dimensional motorized platform for sample positioning [45]. In brief, the ablation was created using a quadruple Nd:YAG laser pulses of 1064 nm. The pulse duration was 8 ns, the pulse energy 1 mJ, and the repetition rate 100 Hz. During the sample scan, the objective to sample distance was carefully controlled to compensate for any flatness anomalies. The light emitted by the plasma was collected and connected by an optical fiber to a Czerny-Turner spectrometer equipped with an intensified charge-coupled device camera (Shamrock 303 and iStar, Andor Technology, Belfast, UK). The experiments were performed by Ablatom S.A.S.

### 2.7. In Ovo Fluorescence Imaging

The experiments were performed at the OPTIMAL (Small animal optical imaging) platform, IAB Grenoble. In order to determine the in ovo tumoral accumulation kinetics, in ovo glioma models were prepared. Eggs bearing well-developed and vascularized glioma xenografted tumors were incubated with 20 µL of aza-SWIR-BSH-01 compound. Fluorescence images of the eggs were acquired before and at specific time intervals following the incubation. Obtained images were analyzed using ImageJ software. Semi-quantitative data regarding the tumoral accumulation were obtained by drawing regions of interest (ROI) around the tumors. NIR-I 2D-fluorescence reflectance imaging device (Fluobeam 800^®^, Fluoptics, France) was used to image the eggs incubated with aza-SWIR-BSH-01 compound. Fluobeam 800 was provided by a class 1 expanded laser source at 780 nm delivering 10 mW/cm^2^ on the imaging field. The resulting fluorescence signals were collected by a CCD through a high pass filter > 830 nm.

### 2.8. Quantification of B Content by Inductively Coupled Plasma—Atomic Emission Spectrometry (ICP-AES)

Mice bearing U-87 MG tumors were injected with 40 µg of ^10^B equivalent from aza-SWIR-BSH-01 or ^10^B-BSH condition (*n* = 3/group). Determination of the boron content in the samples was performed by ICP-AES analyses (Agilent 720 ES) with a detection limit of 0.1 mg/L. The samples were mineralized using 1 mL of aqua regia (mixture of acids: nitric and hydrochloric). After complete mineralization, the samples were diluted with HNO_3_ (5%, *w*/*w*) to reach a 3 mL volume and finally filtered at 0.2 µm for the measurements. The results were expressed as µg of ^10^B/g of tumor. The experiments were performed at the ISTerre platform, Grenoble.

## 3. Results

### 3.1. Rationale, Design, and Characterization of the Compounds

To design an efficient theranostic boron vector, we used a fluorescent reporter platform based on the versatile BODIPY family (Figure 1a) displaying (i) a strong tumor accumulation and (ii) a biocompatible water-solubility, (iii) which can be easily functionalized. In particular, we used *B*-functionalized aza-BODIPYs (Figure 1b,c), as they have numerous advantages, such as a very high chemical and photochemical photostability, and excellent photophysical properties, rendering them suitable fluorophore for the NIR-I to the SWIR region (Figure 1d). Previously, we developed a strategy, which favored their solubilization and limited their aggregation in biologically relevant media (i.e., water, PBS, or serum), by substituting the fluorine atoms on the boron by alkyne ammonium groups [34,35,48]. Very recently, we used this strategy on a particular Donor-Acceptor-Donor’ aza-BODIPY structure, which emits in the SWIR region, yielding a water-soluble derivative SWIR-WAZABY-01 (See the ESI and Reference [29]; *WAZABY* for Water-Soluble aza-BODIPY). SWIR-WAZABY-01 could accumulate in vivo very efficiently at the tumor site, without any vectorization, showing high SWIR contrast for up to one week [29].

In this study, we took advantage of the SWIR-WAZABY-01 structure to deliver the ^10^B-BSH at the tumor site, while enabling the tracking of the molecule in vivo by optical imaging. In order to tether the ^10^B-BSH on the SWIR-WAZABY-01, we slightly modified its structure and adapted the synthetic method previously reported by us in order to synthesize aza-SWIR-BSH-01. It is functionalized with one ^10^B BSH unit and emits in the near-infrared region (Figure 1c and see ESI section for the synthetic pathway of aza-SWIR-BSH-01).

### 3.2. In Vitro Distribution and BNCT Efficacy

To investigate the tumor cell accumulation capacity of aza-SWIR-BSH-01 compounds, the first set of experiments was conducted on brain tumor cells U-251 MG and U-87 MG using SWIR-WAZABY-01, a SWIR emitting aza-BODIPY structurally close to aza-SWIR-BSH-01 but without BSH (see Godard et al. [29]). While the fluorescent signal of SWIR-WAZABY-01 was more compatible with in vivo experiments, due to its emission in the SWIR optical region, it was still detectable by conventional fluorescence microscopy for in vitro investigations. SWIR-WAZABY-01 was incubated with human brain tumor cells U-251 MG and U-87 MG cultured in 2D, but also in 3D spheroids mimicking small tumors. This compound displayed a very fast tumor cell internalization in both 2D and 3D cultures.

Therefore, similar experiments were conducted using the new compound aza-SWIR-BSH-01 on U-87 MG cells. As displayed in Figure 2 after 3 h of incubation, aza-SWIR-BSH-01 accumulated massively in tumor cells, into small cytoplasmic vesicles. The main challenge of this compound was its fluorescence emission properties that can mostly be measured after 800 nm. A dedicated confocal setting was used to collect with a high sensitivity to the photon of the compound. In the first hours of incubation, the signal of the internalized aza-SWIR-BSH-01 was weak and diffuse, but it was clearly visible at around 3 h of incubation as demonstrated in Figure 2a. Under the same experimental conditions, the control cells presented a very weak and homogenous signal (Figure 2b).

To confirm cellular uptake and to further evaluate the compound’s therapeutic potential in vitro, the cells were incubated with aza-SWIR-BSH-01 or ^10^B-BSH for 2 h, placed in quartz cuvettes, and exposed to the neutron beam for 1 to 10 min, or kept as control. The cells were then harvested and seeded for colony assay. After incubation with ^10^B-BSH and short neutron exposure (5 and 10 min), only the growth of U-87 MG cells was reduced as compared to the neutron exposure alone (Figure 3a). Similarly, aza-SWIR-BSH-01 with neutron irradiation was able to strongly reduce the number of colonies as compared to neutron alone (Figure 3a). Regarding U-251 MG cells (Figure 3b), aza-SWIR-BSH-01 strongly decreased the number of colonies as compared to both the ^10^B-BSH and neutron condition after 5 and 10 min neutron exposure (see also ESI Appendix A). These results confirmed the therapeutic potential of aza-SWIR-BSH-01 in vitro, which is slightly better than ^10^B-BSH alone.

### 3.3. In Vivo Distribution and Behavior

Before studying the therapeutic potential of aza-SWIR-BSH-01 compound as a boron-vector, in vivo experiments were performed to determine the distribution profile of the compounds, and more importantly the tumor and tumor environment’s uptake. The animals were imaged before and until 48 h post-intravenous administration of the compound using optical imaging, and the organs were collected at 24 and 48 h post-injection. As indicated in Figure 4a,b, the compound was largely distributed in the animal’s body without any unexpected accumulation and was mainly eliminated by the kidneys. Aza-SWIR-BSH-01 accumulated very weakly in muscles, fat, and healthy brain tissues. Contrariwise, aza-SWIR-BSH-01 accumulated strongly in U-87 MG tumors and was retained for a prolonged time. The remoted tumors revealed a homogenous distribution of aza-SWIR-BSH-01 at 24 and 48 h post-injection (Figure 4c). In parallel, dedicated quantitative experiments were performed in mice-bearing U-87 MG injected with aza-SWIR-BSH-01 and ^10^B-BSH to determine the boron tumor uptake at 24 h. Mice were injected intravenously with either 200 µL of aza-SWIR-BSH-01 at 2 mM or ^10^B-BSH equivalent, i.e., 40 µg of ^10^B equivalent per injection (i.e., >3-fold the dose administered for imaging purpose). Such a concentration of aza-SWIR-BSH-01 is sub-optimal for BNCT experiment, but still allows the optical tracking of the compound. Indeed, the fluorescent compound becomes partially quenched at high dose, which limits its detection. The ICP-AES revealed a tumor uptake of 9.3 ± 4.7 µg of ^10^B/g of tissue and 6.3 ± 4.7 µg of ^10^B/g of tissue, respectively. However, this difference was not significant (*n* = 3/condition). The Tumor/Skin ratios were determined ex vivo; they reached 1.9 ± 0.5 at 24 h, and then decreased at 48 h at 1.5 ± 0.7 (Figure 4d). In Figure 4e, the ex vivo Tumor/Muscle ratios measured at 24 and 48 h were 7.3 ± 0.9 and 12.3 ± 4.4, respectively. Such Tumor/Muscle ratios were in accordance with the required values for BNCT purpose (i.e., T/M ratio > 3–5) [4,7,51]. Furthermore, the investigations also revealed the absence of acute toxicity after intravenous administration of aza-SWIR-BSH-01 [29].

### 3.4. In Ovo BNCT Assay and Distribution

A specific model of tumor growth was used to perform the evaluation of the BNCT efficacy of aza-SWIR-BSH-01 compound. We worked with CAM into which U-251 MG or U-87 MG tumor cells were implanted (Figure 5a). To perform the BNCT experiment, aza-SWIR-BSH-01 and ^10^B-BSH (1.35 µg ^10^B/egg) were added on the top of the U-251 MG tumors 1 h before neutron exposure (Figure 5b). The tumors were collected at day-16, i.e., few days before hatching. The tumor growth was significantly lower in the condition aza-SWIR-BSH-01 + neutron exposure as compared to neutron exposure alone, or ^10^B-BSH + neutron exposure, with 58.4 ± 16.2% versus 100 ± 37.6%, and 101.1 ± 16.0%, respectively (Figure 5c).

We verified the presence of boron at the tumor site for boron-containing conditions to better understand the obtained results. Thus, the excised tumors collected at day 16 were sliced and analyzed by LIBS imaging for elemental analysis [44] and examine phosphorus (P) and boron (B) content and distribution. P was selected as it is present in every cell and its distribution reflects the area of the tissue itself [47,52]. P was used to delineate the tissue sections (in white in Figure 5d). As indicated in Figure 5d, B was observed in tumors treated with aza-SWIR-BSH-01, in all the different sections, while it was not observed for the ^10^B-BSH condition. In this case, ^10^B-BSH may accumulate at the tumor site, but presumably to a lower extent, and/or might be released from the tumor site after such a long time, being below the limits of detection of the LIBS system. Contrariwise, aza-SWIR-BSH-01 accumulated at very high levels in the tumor region (Figure 5d) and was retained in the tissues during 6 days after administration.

From the elemental images obtained with LIBS, we found that the distribution of boron atoms was not homogenous within tumors. A detailed fluorescence distribution study was then conducted in eggs bearing tumors to further understand the compound’s tumor uptake in ovo. The results are presented in Figure 5e,f. Aza-SWIR-BSH-01 accumulated very well in the tumor at early time points (1 h), evidencing some diffusion into vessels at time points with prolonged exposure, suggesting a long-lasting circulation in the blood vessels. The fluorescence signal increased with time, reaching a massive uptake peak at 24 h. As compared to the fluorescence signal obtained after 1 h, the fluorescent signal at 24 h was enhanced by a factor of ~2.5, confirming that the optimal and minimal time required for optimal aza-SWIR-BSH-01 tumor uptake was 24 h. These results also suggest that 24 h post-treatment could be the ideal delay for starting neutron irradiation for a more effective BNCT.

## 4. Discussion

BODIPY derivatives have already been developed for BNCT application [38,39,40,41], but none of them, to our best knowledge, have been tested for BNCT anti-tumor development in vitro and in vivo. This study is the first to report the use and evaluation of aza-BODIPY-based compounds as theranostic compounds for BNCT purpose.

The delivery of ^10^B into tumors for BNCT requires the production and evaluation of biocompatible and water-soluble vectors, with no or only marginal and reversible toxicity, and preferential tumor accumulation while clearing from the blood and tumor micro-environment. As recently reported, some SWIR-WAZABY compounds possess a strong, time-dependent tumor uptake, and weak muscle accumulation [29]. Contrariwise, ^10^B-BSH, a molecule approved for BNCT, is rapidly washed out from the body and should be administrated at high dose [8,53]. Here, the tumor uptake kinetics and the elemental imaging indicated that the ^10^B-BSH part of the novel aza-SWIR-BSH compound advantageously followed the BNCT-compatible kinetics of the SWIR-WAZABY vector, rather than that of the ^10^B-BSH moiety itself. Therefore, the engraftment of the small ^10^B-BSH entity did not impact the tumor accumulation capacity and water-solubility of the SWIR-WAZABY, even when administered at high concentrations (200 µL at 2 mM as a bolus intravenous administration). The substitution of the fluorine atoms on the boron allowing to increase the water-solubility is a key factor for efficient distribution and tumor targeting.

To evaluate the properties of the theranostic aza-SWIR-BSH compound in vivo, the CAM model was used. This model is particularly interesting for drug evaluation as (i) the tumors grow rapidly due to the nutrients-rich environment of the CAM, (ii) the vasculature is well developed and accessible, allowing an efficient tumor vascularization, (iii) the absence of a well-established immune system at this point of embryo development contributes to the tumor growth, even from patient-derived cells, and (iv) the utilization of this experimental model contributes to the reduction of animal’s use in an ethical aspect of the research [54]. Therefore, this model has been widely used for the evaluation of anti-angiogenic or antitumor compounds [54,55,56], the evaluation of sensitizers for radiotherapy and photodynamic therapy [57,58], and imaging using MRI and PET/CT [59,60]; however, the CAM model has never been reported to demonstrate the potential of boron compounds for BNCT application. Using optical imaging, the distribution of aza-SWIR-BSH compound in this CAM model indicated an optimized tumor uptake after prolonged exposure time (24 h, Figure 5), while the 1-h administration followed by neutron exposure was already able to reduce tumor growth; all these results suggested that BNCT could increase tumor shrinkage after prolonged incubations with these compounds and neutron exposure.

## 5. Conclusions

Altogether, we demonstrated that water-soluble aza-BODIPYs can be used as theranostic vectors for boron complexes, opening a new perspective for compound development for BNCT applications.

## 6. Patents

To be added during the reviewing process when official patent numbers will be known.

## Figures and Tables

**Figure 1 cells-09-01953-f001:**
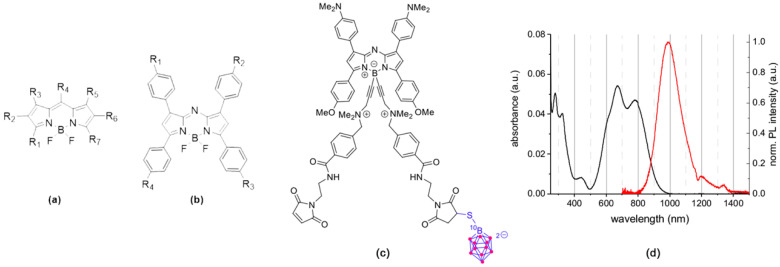
(**a**) General structure of boron-dipyrromethene dyes (BODIPY), (**b**) aza-BODIPY, and (**c**) aza-SWIR-BSH-01, (**d**) absorption and emission spectra of aza-SWIR-BSH-01 in DMSO [29].

**Figure 2 cells-09-01953-f002:**
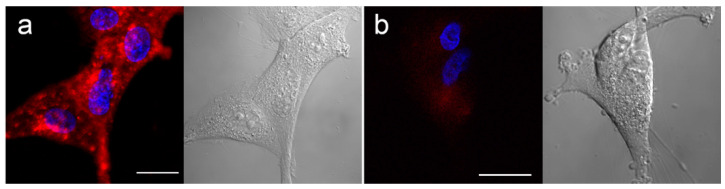
Human glioblastoma U-87 MG cells cultured in 2D incubated with aza-SWIR-BSH-01 (red signal) for 3 h (**a**), and control cells (**b**). Nuclei were labeled with Hoechst (blue signal). Corresponding phase-contrast pictures are depicted in black and white color. Scale bars represent 20 µm.

**Figure 3 cells-09-01953-f003:**
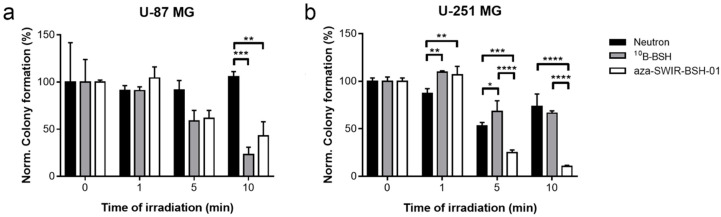
Boron neutron capture therapy (BNCT) experiment in vitro. U-87 MG (**a**) and U-251 MG (**b**) cells were incubated with ^10^B-BSH (grey) or aza-SWIR-BSH-01 (white) during 2 h before neutron exposure, and re-seeding for colony assay. Control condition (neutron alone) is indicated in black. The results are represented as the mean of 3 independent experiments ± standard deviation (S.D.).

**Figure 4 cells-09-01953-f004:**
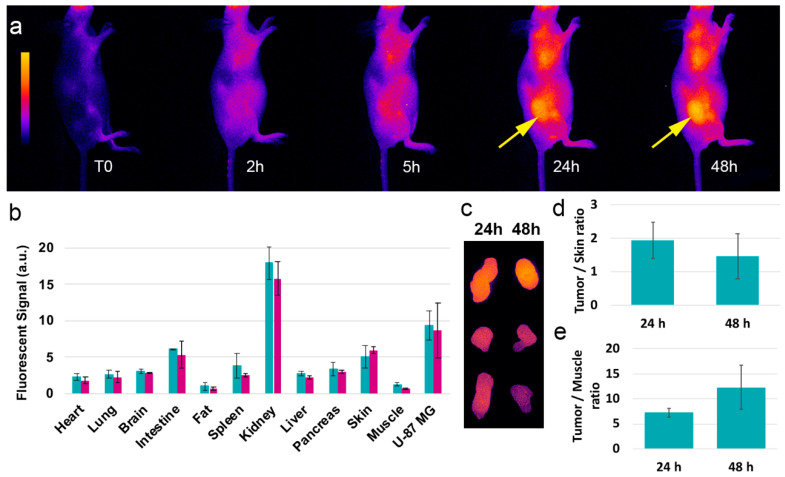
In vivo distribution and behavior of aza-SWIR-BSH-01 in mice-bearing subcutaneous U-87 MG tumors. The non-invasive images were taken from T0 until 48 h (**a**). Tumors are indicated with an arrow. (**b**) The distributions were observed at 24 h (green) and 48 h (blue) post-injection. (**c**) Remoted tumor observed at 24 h and 48 h post-injection revealed a higher tumor accumulation at 24 h. (**d**) Tumor/Skin and (**e**) Tumor/Muscle ratios from ex vivo analysis.

**Figure 5 cells-09-01953-f005:**
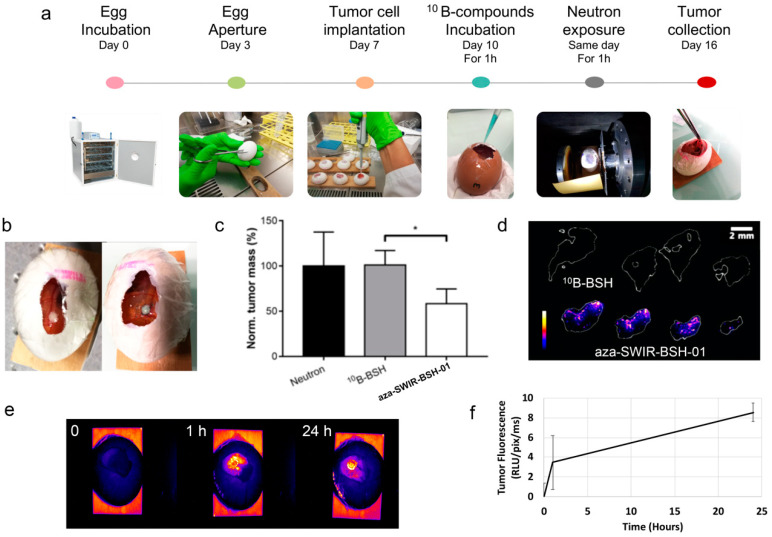
In ovo model of tumor for evaluation of theranostic aza-SWIR-BSH-01 compound. (**a**) Presentation of the in ovo tumor model for BNCT application. (**b**) Color images of tumor before and after administration of aza-SWIR-BSH-01 (pale blue color), indicating the presence of the compound at the tumor site. (**c**) Tumor development measured at day 16, i.e., 6 days after the addition of ^10^B-BSH and aza-SWIR-BSH-01 for 1 h followed by neutron exposure. (**d**) Laser-induced breakdown spectroscopy (LIBS) elemental imaging of boron from tumor sections collected at day 16 showing the presence of remaining boron in tumors treated with aza-SWIR-BSH-01. (**e**) 2D fluorescence imaging of aza-SWIR-BSH-01 distribution before and until 24 h post-administration onto glioma tumors implanted on chorioallantoic membrane (CAM). (**f**) Non-invasive measurement of aza-SWIR-BSH-01 fluorescence in tumors with time. Results are expressed as the tumor fluorescence mean ± S.D (*n* = 4).

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
