# Peer review of "Aza-BODIPY: A New Vector for Enhanced Theranostic Boron Neutron Capture Therapy Applications"

_cells, 2020, doi:10.3390/cells9091953_

Round 1

Reviewer 1 Report

The manuscript by Kalot et al describes the synthesis of an aza-BODIPY conjugated to a BSH boron cluster and an extensive characterization both in vitro and in vivo. The work is extremely interesting as the authors address some of the most relevant challenges for the development of efficient boron vectors for BNCT. Of particular interest is the use of a theranostic derivative that allows the monitoring of the molecule in vitro and in vivo. Moreover, the biological characterization is extensive and exploits many different and innovative methods. Finally, the work is clearly presented and well supported by experimental data.

Therefore the manuscript can be accepted for publication with some minor modifications reported below:

  • On page 2, lines 55-56 the authors state that BSH and BPA are the two compounds clinically approved; however, although not used for a long time, also GB-10 ([closo-B10H10]2−, dodecahydrododecaborate) is an FDA approved molecule see e. g. Journal of Neuro-Oncology 2003, 62:, 33–45.
  • In the manuscript are extensively discussed the advantages of the use of NIR-II/SWIR BODIPY emitters; however it is unclear the penetration depth these compounds allow. Can the authors add some information on this respect?
  • In Supplementary information is described the synthesis of aza-SWIR-BSH-01. In the last conjugation step the procedure describes the addition reaction of BSH on the doubly maleimido functionalized aza-BODIPY using two equivalents of BSH. Careful examination of the NMR spectrum allows to see that the signal at about 7 ppm, attributable to the maleimide protons, integrates for 1.5 protons while it should be 2. Moreover, looking at the whole spectrum, one can argue the presence of a mixture of two very similar compounds. I presume that the aza-SWIR-BSH-01 may be contaminated by a doubly substituted derivative. Please add some information on the purity of aza-SWIR-BSH-01.
  • On page 8, lines 311-313 the following sentence is not fully clear: “The ICP-AES revealed an uptake of 9.3 ± 4.7 μg of 10B/g of tissue, and 6.3 ± 4.7 μg of 10B/g of tissue, respectively.” I presume that the uptake refers to tumor tissue. Moreover, the data reveals that the uptake is below the theoretical limit of 20 μg 10B/g of tumor tissue. Please add some more information.
  • On page 8, Figure 4b the colors of the two bars are very similar, it would be better to use two more different colors. This is just an aesthetic suggestion, feel free to leave the current colors if you prefer.

Author Response

Reviewer #1

The manuscript by Kalot et al describes the synthesis of an aza-BODIPY conjugated to a BSH boron cluster and an extensive characterization both in vitro and in vivo. The work is extremely interesting as the authors address some of the most relevant challenges for the development of efficient boron vectors for BNCT. Of particular interest is the use of a theranostic derivative that allows the monitoring of the molecule in vitro and in vivo. Moreover, the biological characterization is extensive and exploits many different and innovative methods. Finally, the work is clearly presented and well supported by experimental data.

Therefore the manuscript can be accepted for publication with some minor modifications reported below:

We thank Reviewer #1 for critically reviewing our manuscript and for the positive feedback on this study. We addressed each of his/her points in the following section.

  • On page 2, lines 55-56 the authors state that BSH and BPA are the two compounds clinically approved; however, although not used for a long time, also GB-10 ([closo-B10H10]2−, dodecahydrododecaborate) is an FDA approved molecule see e. g. Journal of Neuro-Oncology 2003, 62:, 33–45.

We have highlighted the 2 clinically most used compounds in the clinic and added GB-10 in the new version of the manuscript.

Moreover, two main compounds are clinically approved as 10B-sources: sodium mercaptoundecahydrododecaborate (also called sodium borocaptate, Na210B12H11SH; Na210BSH) and L-p-boronophenylalanine (L-10BPA). While not used in clinical trials, GB-10 ([closo-B10H10]2−, dodecahydrododecaborate) is also an FDA approved molecule.[14]

  • In the manuscript are extensively discussed the advantages of the use of NIR-II/SWIR BODIPY emitters; however it is unclear the penetration depth these compounds allow. Can the authors add some information on this respect?

The penetration depth of NIR-II fluorophores is higher than for NIR-I fluorophores and even higher than “visible-emitting fluorophores”. In qualitative terms, visible-emitting fluorophore are suitable for imaging cells. Because of the higher penetration depth of NIR-I fluorophores, studies with small animals (ex vivo and in vivo imaging) can be performed. NIR-II probes enable to obtain information from deeper tissues, or to increase the resolution of the images as compared to NIR-I probes.

Penetration depth depends on the nature of the tissues, the optical properties of the probe itself, the probe’s concentration and more importantly, the excitation and the signal collection system. In addition, some specific signal processing can strongly increase the quality of the images. However, some information for specific signal collected from deep tissues might be found in Musnier et al Nanoscale. 2019 Jul 7;11(25):12092-12096, and in the recent Zu Z. et al ACS Nano 2020 DOI: 10.1021/ acsnano.0c01174.

The essential point with NIR-II optical window is that it enables more precise diagnosis and prognosis in deep tissues, with reduced auto-fluorescence, decreased photo-scattering, and less interfering absorption than the visible and NIR-I channels. All the studies reporting NIR-II imaging confirmed these observations.

Fluorophores displaying close photophysical properties enable imaging between one to ten millimeters of penetration depth. One of them was even able to enable a penetration depth of 30 mm in chicken tissue (see summary table in Journal of Luminescence 225 (2020) 117338 / doi: 10.1016/ j.jlumin.2020.117338).

Using our settings and conditions, compatible with repeated imaging and biological concentrations, the probe could be visualized within animal tissues without specific signal processing in the first 3-4 mm.

The following sentence has been added to the manuscript: “SWIR fluorophores enable to reach a penetration depth of 1 to 10 mm with increased resolution of the images as compared to those obtained with NIR-I probes [42,43].”

  • In Supplementary information is described the synthesis of aza-SWIR-BSH-01. In the last conjugation step the procedure describes the addition reaction of BSH on the doubly maleimido functionalized aza-BODIPY using two equivalents of BSH. Careful examination of the NMR spectrum allows to see that the signal at about 7 ppm, attributable to the maleimide protons, integrates for 1.5 protons while it should be 2. Moreover, looking at the whole spectrum, one can argue the presence of a mixture of two very similar compounds. I presume that the aza-SWIR-BSH-01 may be contaminated by a doubly substituted derivative. Please add some information on the purity of aza-SWIR-BSH-01.

We fully understand the reviewer’s remark. Indeed, we also believed that a double addition of BSH can occur. However, even in presence of a large excess of BSH and stirring at reflux, we never succeeded for the moment in introducing 2 BSH moieties. Moreover, when we monitored the reaction by HPLC-MS, no trace of doubly substituted derivative was observed (see Figure REV1 below).

PLEASE SEE THE ATTACHMENT

Figure REV1: Analytical HPLC for aza-SWIR-BSH-01 reaction monitoring

  • On page 8, lines 311-313 the following sentence is not fully clear: “The ICP-AES revealed an uptake of 9.3 ± 4.7 μg of 10B/g of tissue, and 6.3 ± 4.7 μg of 10B/g of tissue, respectively.” I presume that the uptake refers to tumor tissue. Moreover, the data reveals that the uptake is below the theoretical limit of 20 μg 10B/g of tumor tissue. Please add some more information.

The dose administrated for this experiment was selected to mimic an experiment of BNCT in sub-optimal concentrations, while still allowing the detection of the compound. Indeed, the compound becomes partially quenched at high dose which limits its detection. Therefore, we make a compromise between optical imaging and tumor uptake for BNCT experiment.

We apologize for the lack of accuracy and modified the sentences as follow:

Such a concentration of aza-SWIR-BSH-01 is sub-optimal for BNCT experiment, but still allows the optical tracking of the compound. Indeed, the fluorescent compound becomes partially quenched at high dose which limits its detection. The ICP-AES revealed a tumor uptake of 9.3 ± 4.7 µg of 10B/g of tissue, and 6.3 ± 4.7 µg of 10B/g of tissue, respectively.”

  • On page 8, Figure 4b the colors of the two bars are very similar, it would be better to use two more different colors. This is just an aesthetic suggestion, feel free to leave the current colors if you prefer.

The colors have been changed accordingly.

Reviewer 2 Report

The manuscript by Kalot et al. described the synthesis and biological evaluation of the fluorescent aza-BODIPY/10B-BSH compound, aza-SWIR-BSH-01, as a new vector for enhanced theranostic agent for boron neutron capture therapy. The authors previously reported water-soluble aza-BODIPYs which possess high contrast in vivo NIR-II imaging. In the current study, the authors achieved selective delivery of aza-SWIR-BSH-01 to U-87 MG tumor in vivo. The results are promising and this reviewer strongly recommend the manuscript suitable for publication in Cells.

However, the following issue should be revised before publication:

1) The boron uptakes are described in lines 308-313. However, the subscription is not clear for readers to understand. What tissues do these concentrations represent? The authors are strongly requested to provide clear explanation in this regard.

2) Why did aza-SWIR-BSH-01 selectively accumulate into tumor? The authors should address the uptake mechanism. This is important for readers to understand the novelty of this study.

3) aza-SWIR-BSH-01 has another reactive moiety with BSH. Is it possible to introduce another BSH into aza-SWIR-BSH-01? Why didn’t the authors synthesize bis-BSH compound and demonstrate biological activity?

4) The abbreviation “NIR-I” and “SWIR” in line 73 should be explained at the same line instead of line 85.

Author Response

Reviewer#2

The manuscript by Kalot et al. described the synthesis and biological evaluation of the fluorescent aza-BODIPY/10B-BSH compound, aza-SWIR-BSH-01, as a new vector for enhanced theranostic agent for boron neutron capture therapy. The authors previously reported water-soluble aza-BODIPYs which possess high contrast in vivo NIR-II imaging. In the current study, the authors achieved selective delivery of aza-SWIR-BSH-01 to U-87 MG tumor in vivo. The results are promising and this reviewer strongly recommend the manuscript suitable for publication in Cells.

We thank Reviewer #2 for critically reviewing our manuscript and for the positive feedback on this study. We addressed each of his/her points in the following section.

However, the following issue should be revised before publication:

1) The boron uptakes are described in lines 308-313. However, the subscription is not clear for readers to understand. What tissues do these concentrations represent? The authors are strongly requested to provide clear explanation in this regard.

We apologize for the lack of precision. The new sentence (line 315 of the current version) is: “The ICP-AES revealed a tumor uptake of 9.3 ± 4.7 µg of 10B/g of tissue, and 6.3 ± 4.7 µg of 10B/g of tissue, respectively”.

2) Why did aza-SWIR-BSH-01 selectively accumulate into tumor? The authors should address the uptake mechanism. This is important for readers to understand the novelty of this study.

The tumor uptake of aza-SWIR-BSH-01 occurs via passive EPR effect (enhanced permeability and retention effect). This is not better indicated in the manuscript (line 90): “We recently reported the use of a B-functionalized aza-BODIPY, emitting in the SWIR region and accumulating in tumors via EPR effect (Enhanced permeability and retention effect) with high and prolonged Tumor/Healthy Muscle ratio (between 5 and 30, from 24 to 168 hours post-injection) without the need of tethering vector [29]".

3) aza-SWIR-BSH-01 has another reactive moiety with BSH. Is it possible to introduce another BSH into aza-SWIR-BSH-01? Why didn’t the authors synthesize bis-BSH compound and demonstrate biological activity?

We agree with the reviewer. It could be interesting to synthesize the bis-BSH derivative. However, even in presence of a large excess of BSH and stirring at reflux, we never succeeded in introducing two BSH moieties for the moment. We believed that both the two minus charges of the BSH and the steric hindrance prevent the second equivalent of BSH to approach, and thus, to react with the second maleimide group of the aza-BODIPY platform. Therefore, the synthesis of the bis-BSH aza-BODIPY will imply to change the synthetic strategy. For example, we may consider increasing the linker length between the BODIPY core and the maleimide moiety.

4) The abbreviation “NIR-I” and “SWIR” in line 73 should be explained at the same line instead of line 85.

This modification has been done accordingly.

Reviewer 3 Report

The manuscript outlined design and synthesis of an aza-BODIPY derivative: a new vector for enhanced theranostic boron neutron capture therapy applications.  The experimental settings are technically sound and the manuscript was written and organized in well manner. The combination of water soluble aza-Bodipy derivative with clinically approved 10B-BSH for BNCT therapy would be appreciated and the results are interesting. However, the manuscript contains several ambiguous points to be revised. The detailed comments are listed below.

1) Authors stated that “SWIR-WAZABY-01 could accumulate in vivo very efficiently at the tumor site, without any vectorization, showing high SWIR contrast for up to one week’’. Authors may add any explanation what makes the azabodipy derivative to target the tumor and compare their results with othres (RSC advances, 2019, 9, 13372; Analyst, 2019, 144, 2393; ACS appl. Bio Mater. 2020, 3, 1, 45).

2) Authors mentioned that ‘’In the first hours of incubation, the fluorescence signal of aza-SWIR-BSH-01 was very week and it is clearly visible at 3 h’’ (p 6, line 270 and Figure 2). Isn't there any possibility of bond breakage between aza-BODIPY and BSH?

3) In Figure 3, authors compared BNCT effect in two different cell lines and the results are interesting. BSH showed better effect in U-251 MG cells but aza-SWIR-BSH-01 showed better effect in U-87 MG cells. Any explanation?

4) aza-SWIR-BSH-01 contains one free maleimide derivative, there might be a possibility of reaction with intracellular biothiols (H2S, Cysteine and GSH). Is there any reason of conjugation of only one BSH to BODIPY?

5) Some key data are missing.

- what is water solubility of aza-SWIR-BSH-01?

- Measure in vivio stability of aza-SWIR-BSH-01. Although aza-BODIPY is stable, there is no data regarding with statility of aza-SWIR-BSH-01 itself.

- Measure boron concentration in tumor. When there is any beakage of linker between BODIPY and BSH, fluorescence signals do not represent boron contents correctly.

6) In tumor diagnosis experiments, authors used U87MG cells, but in ovo CAM models, U251MG cells were used. Any explanation?

7) Any possibility of BBB penetaration of aza-SWIR-BSH-01?

Author Response

Reviewer#3

The manuscript outlined design and synthesis of an aza-BODIPY derivative: a new vector for enhanced theranostic boron neutron capture therapy applications.  The experimental settings are technically sound and the manuscript was written and organized in well manner. The combination of water soluble aza-Bodipy derivative with clinically approved 10B-BSH for BNCT therapy would be appreciated and the results are interesting. However, the manuscript contains several ambiguous points to be revised. The detailed comments are listed below.

We thank Reviewer #3 for critically reviewing our manuscript and we addressed each of his/her points in the following section.

1) Authors stated that “SWIR-WAZABY-01 could accumulate in vivo very efficiently at the tumor site, without any vectorization, showing high SWIR contrast for up to one week’’. Authors may add any explanation what makes the azabodipy derivative to target the tumor and compare their results with othres (RSC advances, 2019, 9, 13372; Analyst, 2019, 144, 2393; ACS appl. Bio Mater. 2020, 3, 1, 45).

Aza-SWIR-BSH-01, as aza-BODIPY without BSH, passively accumulates in the tumor. The main difference with the other aza-BODIPYs (RSC advances, 2019, 9, 13372; Analyst, 2019, 144, 2393; ACS appl. Bio Mater. 2020, 3, 1, 45) is the boron functionalization for water-solubilization. This modification improves the bioavailability and probably the interactions with other biological components. In the 3 articles cited, none of the aza-BODIPY is clearly water-soluble. Only one of them is administrated intratumorally. Similarly, Bai et al (Chem Commun (Camb) 2019, 55, 10920-10923, doi:10.1039/c9cc03378e) used a vectorization system to administrate the aza-BODIPY for tumor targeting. We think that the water-solubilization is a key part of the reported aza-BODIPY.

A sentence has been added in the discussion to explain this uptake: “The substitution of the fluorine atoms on the boron allowing to increase the water-solubility is a key factor for efficient distribution and tumor targeting.

2) Authors mentioned that ‘’In the first hours of incubation, the fluorescence signal of aza-SWIR-BSH-01 was very week and it is clearly visible at 3 h’’ (p 6, line 270 and Figure 2). Isn't there any possibility of bond breakage between aza-BODIPY and BSH?

The preliminary NMR stability studies revealed that the bond between aza-BODIPY and BSH is robust. Moreover, the quantum yield of fluorescence of aza-SWIR-BSH-01 and its analogue without BSH are in the same range. We hypothesize that the increase of the signal in vitro and in vivo might be due to the change of the micro-environment or interaction with biomolecules which modifies either matrix polarity or matrix viscosity or maybe a combination of both. Any modification of the matrix effect leads to enhanced photoluminescence intensity and quantum yield in the SWIR.

3) In Figure 3, authors compared BNCT effect in two different cell lines and the results are interesting. BSH showed better effect in U-251 MG cells but aza-SWIR-BSH-01 showed better effect in U-87 MG cells. Any explanation?

In Figure 3, BSH (grey bars) shows a slightly better effect on U-87 MG cells as compared to aza-SWIR-BSH-01 (white bars), but the difference is not significant. Both compounds have similar efficacy. In U-251 MG, aza-SWIR-BSH-01 has a significantly better efficacy as compared to BSH. This is why we indicated “These results confirmed the therapeutic potential of aza-SWIR-BSH-01 in vitro, which is slightly better than 10B-BSH alone.”

After 2 hours of incubation, the compounds should be internalized similarly in U-87 MG cells, while aza-SWIR-BSH-01 should be more internalized and/or more retained as compared to BSH in U-251 MG cells. This hypothesis may explain this difference.

4) aza-SWIR-BSH-01 contains one free maleimide derivative, there might be a possibility of reaction with intracellular biothiols (H2S, Cysteine and GSH). Is there any reason of conjugation of only one BSH to BODIPY?

Despite the presence of a large excess of BSH and stirring at reflux, we were not able to introduce a second BSH onto the aza-BODIPY in our conditions. We believed that both the two minus charges of the BSH and the steric hindrance prevent the second equivalent of BSH to approach, and thus, to react with the second maleimide group of the aza-BODIPY platform. However, we agree with the reviewer on the possibility of reaction with intracellular biothiols.

5) Some key data are missing.

- what is water solubility of aza-SWIR-BSH-01?

We observed that solubilization of the probe is higher than 600 µm in PBS, water, NaCl 0.9%, glucose 5% or similar solutions.

The photophysical properties have been evaluated at 10 µM, as well as for the in vitro investigations. Using the confocal imaging system, we never observed any probe aggregation. Figure 2 presents some 2D cells incubated with the probe diluted at 10 µM in cell culture media (in absence of red phenol). In the (a) panel, we observed the absence of probe aggregation in the well, as demonstrate by the dark background of the pictures.

Altogether, these results consistently show that the Aza-BODIPYs are Water-Soluble in this concentration range.

- Measure in vivio stability of aza-SWIR-BSH-01. Although aza-BODIPY is stable, there is no data regarding with statility of aza-SWIR-BSH-01 itself.

Aza-SWIR-BSH-01 can be store at room temperature as a powder for prolonged period. It is also stable in solution (alcohol, acetonitrile, DMSO, water…) for several weeks.

In parallel, the stability of BSH-maleimide link was studied in serum and was stable for several hours (reference 19 of the article, doi:10.1016/j.bmc.2011.01.020.). Moreover, as pickpointed by the reviewer, aza-BODIPY is known to be stable, therefore, aza-SWIR-BSH-01 is very likely to be stable in vivo for several days.

- Measure boron concentration in tumor. When there is any beakage of linker between BODIPY and BSH, fluorescence signals do not represent boron contents correctly.

The boron concentration in tumor is indicated line 315 of the new version of the manuscript: “The ICP-AES revealed a tumor uptake of 9.3 ± 4.7 µg of 10B/g of tissue, and 6.3 ± 4.7 µg of 10B/g of tissue, respectively”.

6) In tumor diagnosis experiments, authors used U87MG cells, but in ovo CAM models, U251MG cells were used. Any explanation?

As indicated in Figure 2, the BNCT efficacy of aza-SWIR-BSH-01 was stronger for U-251 MG cells as compared to U-87 MG cells. Therefore, U-251 MG cells were selected for in ovo experiment of BNCT. In vivo, both cell lines are relevant and might be selected. However, U-87 MG cells are able to grow a faster in vivo as compare to U-251 MG cells. Due to planning organization this year, we selected U-87 MG cells for in vivo experiment.

7) Any possibility of BBB penetaration of aza-SWIR-BSH-01?

In absence of any brain lesion, the penetration of aza-SWIR-BSH-01 in the brain is very weak. When tumors are implanted in the brain, preliminary studies indicated that the aza-BODIPY can reach the tumor site. However, we did not performed experiment with aza-SWIR-BSH-01 currently. The compound should probably reach the tumor if the BBB is broken. In addition, small and controlled administration of mannitol (or similar compound) before aza-SWIR-BSH-01 should favor the tumor brain uptake of aza-SWIR-BSH-01.

Round 2

Reviewer 3 Report

All issues raised during the first review were properly addressed.